# Matrix regulation: a plug-and-tune method for combinatorial regulation in *Saccharomyces cerevisiae*

Xiaolong Teng[1,5], Zibai Wang[1,5], Yueping Zhang[2,5], Binhao Wang[1], Guiping Gong[1], Jinmiao Hu[1], Yifan Zhu[1], Baoyi Peng[1], Junyang Wang[1], James Chen[1], Shuobo Shi [1], Jens Nielsen [1,3,4] & Zihe Liu [1] ✉

Transcriptional fine-tuning of long pathways is complex, even in the extensively applied cell factory *Saccharomyces cerevisiae*. Here, we present Matrix Regulation (MR), a CRISPR-mediated pathway fine-tuning method enabling the construction of $6^8$ gRNA combinations and screening for the optimal expression levels across up to eight genes. We first identify multiple tRNAs with efficient gRNA processing capacities to assemble a gRNA regulatory matrix combinatorially. Then, we expand the target recognition of CRISPR regulation from NGG PAM to NG PAM by characterizing dCas9 variants. To increase the dynamic range of modulation, we test 101 candidate activation domains followed by mutagenesis and screening the best one to further enhance its activation capability in *S. cerevisiae* by 3-fold. The regulations generate combinatorial strain libraries for both the mevalonate pathway and the heme biosynthesis pathway and increase squalene production by 37-fold and heme by 17-fold, respectively, demonstrating the versatility of our method and its applicability in fundamental research.

The ability to fine-tune a pathway with multiple genes remains challenging in the field of life sciences and metabolic engineering[1,2]. Underexpression of rate-limiting enzymes often restricts the metabolic flux of the whole pathway, whereas overexpression of certain pathway enzymes beyond what is needed would cause both accumulated intermediates and a waste of cellular resources. A typical approach is therefore overexpressing all genes in the target pathway, followed by multiple rounds of the design-build-test cycles until the optimal expression level of each enzyme is achieved. This overall process is laborious and time-consuming.

To dial in the correct amount of enzymes in a metabolic pathway, two main strategies based on library screening have been developed. One is directly generating genomic diversity within a set of functional elements, e.g., promoters[3], ribosome binding sites[4], 5' untranslated regions (5' UTR), and terminators[5], which may cause untargeted mutations because of massive genome editing. The other approach is combinatorially assembling diverse regulatory modules, such as transcription factors[6], Kozak variants[7], and constitutive promoters[8,9], which may be restricted by current molecular engineering techniques, such as the amount of DNA that can be transformed into a given host strain or the number of genome integrations that can be multiplexed, etc. Thus, both combinatorial strategies are often accompanied by high-throughput assays, which are unfortunately unavailable for most target molecules. Therefore, a low-cost, facile, and less context-dependent fine-tuning technology is still needed to facilitate the optimization of cell factories and the basic research into the relationships between genotypes and phenotypes.

[1]College of Life Science and Technology, State Key Laboratory of Green Biomanufacturing, Beijing Advanced Innovation Center for Soft Matter Science and Engineering, Beijing University of Chemical Technology, Beijing, China. [2]College of Veterinary Medicine, China Agricultural University, Beijing, China. [3]Department of Biology and Biological Engineering, Chalmers University of Technology, Gothenburg, Sweden. [4]BioInnovation Institute, Copenhagen N, Denmark. [5]These authors contributed equally: Xiaolong Teng, Zibai Wang, Yueping Zhang. ✉e-mail: zihe@mail.buct.edu.cn

With the development of Clustered Regularly Interspaced Short Palindromic Repeats (CRISPR) mediated transcriptional reprogramming[10,11] and advanced gRNA assembly methods[12,13], CRISPR-based combinatorial regulation strategies have been reported[14,15]. Yet, CRISPR-based activations rely on Cas proteins binding to the regions upstream of the 5' UTRs that often composed of AT-rich sequences[16], whereas their PAM sequences, such as NGG for the most commonly used SpCas9 (Cas9 from *Streptococcus pyogenes*)[17], often restrict the available operating sites of a given promoter. Furthermore, the dynamic range of activation is limited by the inefficiency of activation domains (ADs) in yeast[18]. For example, the hybrid tripartite activation domain VP64-p65-Rta (VPR) in *S. cerevisiae* could only upregulate randomly picked genes by 2-fold[18,19], while it enables target gene activation by over 10,000-fold in mammalian cells[20].

Here, we develop Matrix Regulation (MR), a plug-and-tune technology that allows fine-tuning of pathways involving up to eight genes in *S. cerevisiae*. Briefly, to realize the combinatorial assembly of multiple groups of gRNAs and efficient processing of the gRNA arrays, we characterize multiple tRNAs and propose mixed tRNA arrays. To achieve a broadened targeting scope and ensure sufficient gRNA varieties within a given promoter, we assess the efficiency of dCas9 variants and identify dSpCas9-NG with efficient regulation capacities over all NG PAMs in *S. cerevisiae*. To address the limited regulation dynamic range caused by insufficient ADs in *S. cerevisiae*, we characterize 101 candidate activation domains, followed by further enhancing the best one by 3-fold through random mutagenesis and high-throughput screening. Finally, to prove the utility of our toolkit in biological research, we apply MR for optimizing the production of both squalene (without the high-throughput screening method reported) and heme (with the 96-wellplate screening method reported). Briefly, through a single-step assembly and transformation, we generate a combinatorial library with a size of $6^8$, in which all eight genes in the mevalonate (MVA) pathway are up-regulated at six activation levels, respectively. Through random picking of 50 colonies without the help of any phenotype aids, squalene production is enhanced by 37-fold. Moreover, we generate a two-dimensional combinatorial library of the heme biosynthesis pathway with the size of $6^8$, and through random picking of 500 colonies, heme production is enhanced by 17-fold.

## Results

### Hybrid tRNA arrays enabled the construction of combinatorial gRNA array libraries

In this study, we aim to build a combinatorial regulation method with a library size of $6^8$, fine-tuning pathways with up to eight genes, with each gene regulated at six levels based on different gRNA positions. Yet, the current gRNA assembly method needs to be optimized for combinatorial purposes.

For genome engineering in yeast, recombinant plasmids generally have to be transformed into *Escherichia coli* (*E. coli*) for plasmid amplification and verification, which is both time-consuming and, more importantly, results in a considerable loss of combinatorial assembly possibilities. The Lightning GTR-CRISPR method developed for multiplexed genome knockouts which employs a gRNA-tRNA^Gly array can skip the *E. coli* cloning step and be directly transformed into yeast[13]. However, this system cannot yet facilitate *E. coli*-less yeast-based combinatorial regulations. In the Lightning GTR-CRISPR system, only tRNA^Gly is used to slice gRNAs of all gene targets, and the recognition site of the Golden Gate assembly enzymes could only be selected within the 20-base pair (bp) targeting sequences, which in turn imposes constraints on the combinatorial assembly of multiple groups of gRNAs. Moreover, using the same tRNA^Gly for all gRNAs may cause more possibilities of mis-constructed plasmids[13]. Briefly, for each gene to be regulated, the same 76-bp scaffold RNA has to be assembled,

which together with the 71-bp tRNA^Gly would be 147 bp. Whereas in *S. cerevisiae* 100 bp homology arms would cause efficient gap repairs, especially in MR where all parts are directly transformed into yeast rather than preassembled in vitro. Thus, the gRNA-tRNA array needs to be optimized with varied tRNAs to facilitate the assembly of varied targeting sequences of one gene (Fig. 1a).

To facilitate subsequent assembly and enable a compact architecture, we selected seven short tRNAs with four distinct nucleotides at the 3' end, including tRNA^Leu, tRNA^Gln, tRNA^Asp, tRNA^Arg, tRNA^Lys, tRNA^Thr, and tRNA^Ser. Sequences for tRNAs used in this study can be found in Supplementary Data 1. The five-gene gRNA-tRNA array format[13] disrupting *CAN1*, *LYP1*, *TRP2*, *FAA1*, and *ADE2* genes was selected for tRNA characterizations. Based on our previous knowledge, in this format, the transcription efficiency of the *SNR52* promoter becomes insufficient when the number of gRNA transcripts increases to more than five[13]. Therefore, we replaced the last tRNA^Gly with candidate tRNAs (Fig. 1b). The disruption of *ADE2* results in a red phenotype of the cells, allowing for rapid assessment of the splicing efficiency of the tRNA under investigation. The results indicated that all tested tRNAs exhibited comparable gene disruption efficiencies with tRNA^Gly, due to the similar RNA splicing efficiencies of these native tRNAs.

### Identification of dSpCas9-NG with broadened PAM recognitions for combinatorial regulations

The widely used CRISPR-Cas9 system from *S. pyogenes* recognizes the NGG PAM sequence. Meanwhile, activation for the gene of interest is typically achieved by targeting upstream of 5"UTR regions, which tend to be AT-rich[16,21], posing challenges in selecting alternative target sequences for combinatorial regulations. To expand targeting scopes and enhance the magnitude of regulation, SpCas9-NG and the xCas9 with NG PAMs reported in human cells[22,23] and demonstrated in *S. cerevisiae* with editing capacities[24] were tested (Fig. 2a).

To avoid gRNA bias, we first constructed a platform strain, integrated into the genome XII-2 locus[25] with a 140 bp minimal *CYC1* promoter, the *mCherry* gene, and the *ADH1* terminator. We then fixed a 20 bp targeting sequence and replaced its CGG PAM with all other possible 15 NG PAMs, generating in total 16 platform strains for Cas9 variant characterizations. As shown in Fig. 2b, the results suggested that dxCas9-VPR did not exhibit significant regulatory capacities for the other 12 NG PAMs except for NGG PAMs in *S. cerevisiae*, nor did dCas9-VPR, while dSpCas9-NG-VPR exhibited efficient regulatory potencies.

These results also suggested that, under all conditions, the highly effective activator (VPR) in mammalian cells could only achieve a 2-fold activation in *S. cerevisiae*, indicating insufficient combinatorial activation potency that needs optimization, which is consistent with the results previously reported in *S. cerevisiae*[18,19].

### Enhancement of the CRISPR-AD VPR through high-throughput screening

The use of a protein scaffold system that recruits multiple copies of ADs to amplify gene expression signals has been reported[18,26,27]. Yet, this approach substantially increases the size of the dCas-AD protein that may impose a cellular burden and reduce the cell viability during multiplexing[18,28]. Thus, we chose to increase the dynamic range of CRISPR-based activations by optimizing its activation domains.

We started by identifying potent endogenous activators from 101 candidates through the *Saccharomyces* Genome Database (SGD). Specifically, three criteria were taken to identify potential DNA sequences: (1) annotated as transcription factors, activators, or regulators, (2) with no reported functions related to repression or inhibition, (3) for those containing DNA-binding domains, the remaining parts were investigated separately. Details of these 101 candidate activators can be found in Supplementary Data 2.

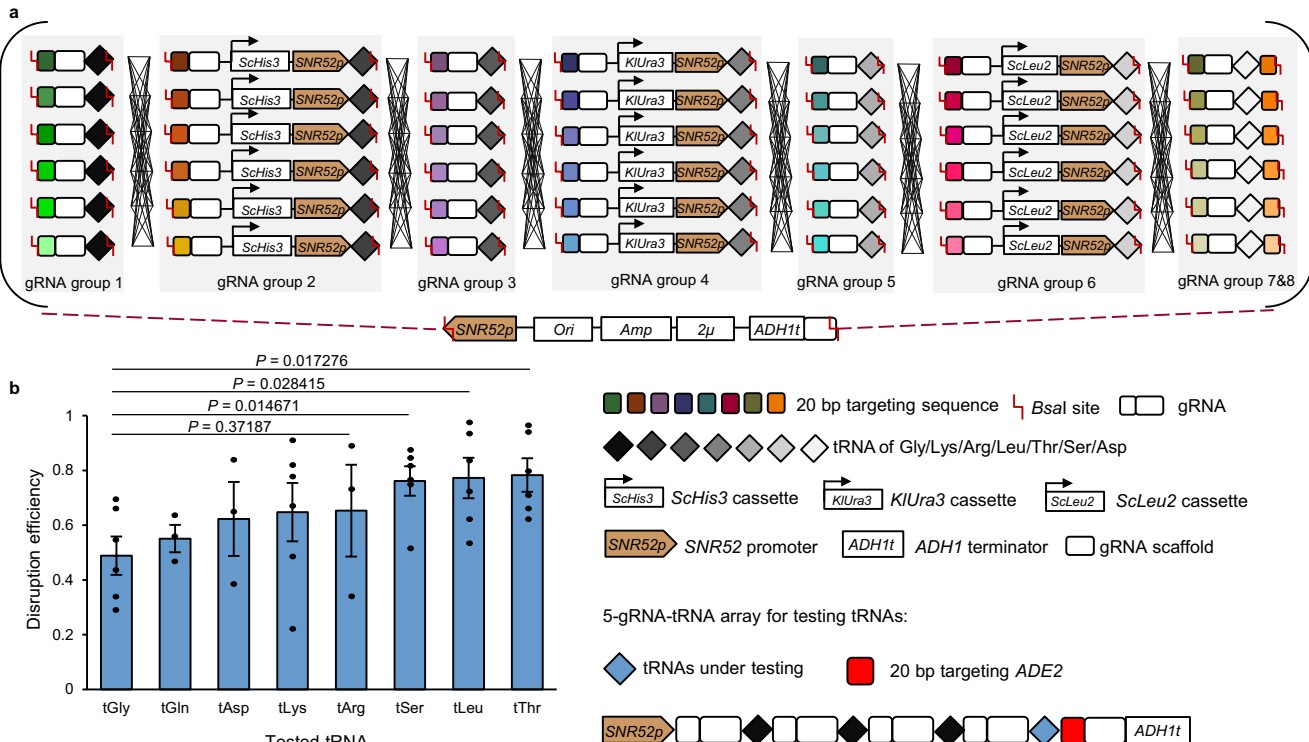

**Fig. 1 | tRNAs identified for combinatorial pathway tuning. a** Graphic representation of the concept of MR. The *dCas9-AD* gene was pre-integrated into the host strain. Plasmid libraries were built using Golden Gate with PCR-generated fragments that included gRNAs, tRNAs, and markers (*S. cerevisiae HIS3*, *Kluyveromyces lactis URA3*, and *S. cerevisiae LEU2*). Then, the Golden Gate mix, including eight groups of gRNAs targeting eight different genes, could be directly transformed into yeast for combinatorial expression and screening. The colored rectangles in a gRNA group represent gRNAs targeting different positions of a gene for regulation. *Ori*, *Amp*, and *2μ* denote the origin of replication, ampicillin resistance gene expression cassette, and 2-micron origin of replication in yeast, respectively. **b** Alternative tRNAs were identified with high RNA-splicing efficiency in the 5-gRNA-tRNA arrays. The abbreviations tGly, tGln, tAsp, tLys, tArg, tSer, tLeu, and tThr represent the selected tRNAs for glycine, glutamine, aspartic acid, lysine, arginine, serine, leucine, and threonine, respectively. The efficiency was calculated by dividing the number of red colonies by the total number of colonies on the plate, and error bars represent mean ± SD of all replicates ($n = 6$). Statistical analysis was performed using two-tailed Student's t-test (*$p < 0.05$). Source data are provided as a Source Data file.

The activation efficiencies were tested based on their abilities to activate the Zeocin resistance gene *ZeoR* through spot growth assays (Supplementary Fig. 1). Specially, potential candidates from Taf4, Pdr1, Snf12, and Med2 exhibited strong activation efficiencies. Taf4 forms part of the TFIID complex involved in RNA polymerase II transcription initiation[29]. Pdr1 is a zinc cluster-containing transcription factor (TF) that regulates the pleiotropic drug response[30]. Snf12 is a subunit of the SWI/SNF chromatin remodeling complex involved in the transcriptional regulation of 5% of the genes in yeast[31]. These three domains have not yet been reported as CRISPR ADs. Yet, results showed that all tested endogenous activators exhibited weaker *ZeoR* activation capabilities compared with VPR, which may be due to the complexity of in vivo regulatory mechanisms suppressing their effectiveness as CRISPR ADs[32]. In contrast to our finding, Med2, a subunit of the RNA polymerase II mediator, slightly outperformed VPR as a CRISPR AD in a recent report[18].

Next, we sought to enhance the commonly used activation domain VPR through random mutagenesis and screening, activating both *mCherry* and *ZeoR* using the same gRNA targeting *CYC1p*. Briefly, the transformants were screened by the Fluorescence-Activated Cell Sorting (FACS), and cells exhibiting intense mCherry signals were sorted into 96-well plates with Zeocin. Then, the more efficient VPR variants were identified by spot growth assay and real-time qPCR (Fig. 3a).

Considering that the dCas9-VPR activated *ZeoR* expression could already endow cell growth on plates with 800 mg/L Zeocin (Supplementary Fig. 1), while Zeocin concentration higher than this level caused false positive growth, we first attenuated the activity of ZeoR

protein. Briefly, the secondary structure of ZeoR protein was predicted using Phyre2, followed by alanine substitutions introduced at 13 conserved charged sites (Fig. 3b). Next, spot growth assays were performed and identified mut8 and mut13 with weak resistance to 50 mg/L of Zeocin (Fig. 3c), while upon VPR activation mut13 exhibited an improved resistance to 100 mg/L of Zeocin (Fig. 3d). Then, during FACS sorting, cells with the VPR activation group displayed increased mCherry signals and could grow at Zeocin concentrations up to 100 mg/L, while cells without VPR activation could proliferate only at concentrations up to 50 mg/L (Fig. 3e). This result also suggested that positive VPR mutants could be isolated using high-throughput screening.

A total of three consecutive rounds of FACS screening were carried out, followed by selections in 96-well plates and spot growth assays with increased Zeocin concentrations (Fig. 3f). Plates without Zeocin were used as growth control. For the initial round, cells with the top 5% mCherry signal were sorted to plates with 200 mg/L or 300 mg/L Zeocin, leading growth of 22 colonies (0–2 for each plate) after three days of cultivation. Spot growth on Zeocin of these colonies indicated that the VPR mutant 13 (named e13), was the most potent mutant after Round 1 (Supplementary Fig. 3). To avoid potential genome mutation effects, e13 together with the other seven variants were verified by reconstructing the mutated VPR plasmids and retransforming them into the starting strain with a clean background (Supplementary Fig. 5). Similarly, mutants e137 and e13711 were successively obtained from Round 2 and Round 3, respectively (Supplementary Fig. 4). In total, $1.5 \times 10^{8}$ yeast cells (three rounds of FACS sorting were performed, each using 1 mL of cell culture at an $OD_{600}$ of 5, and each $OD_{600}$ was

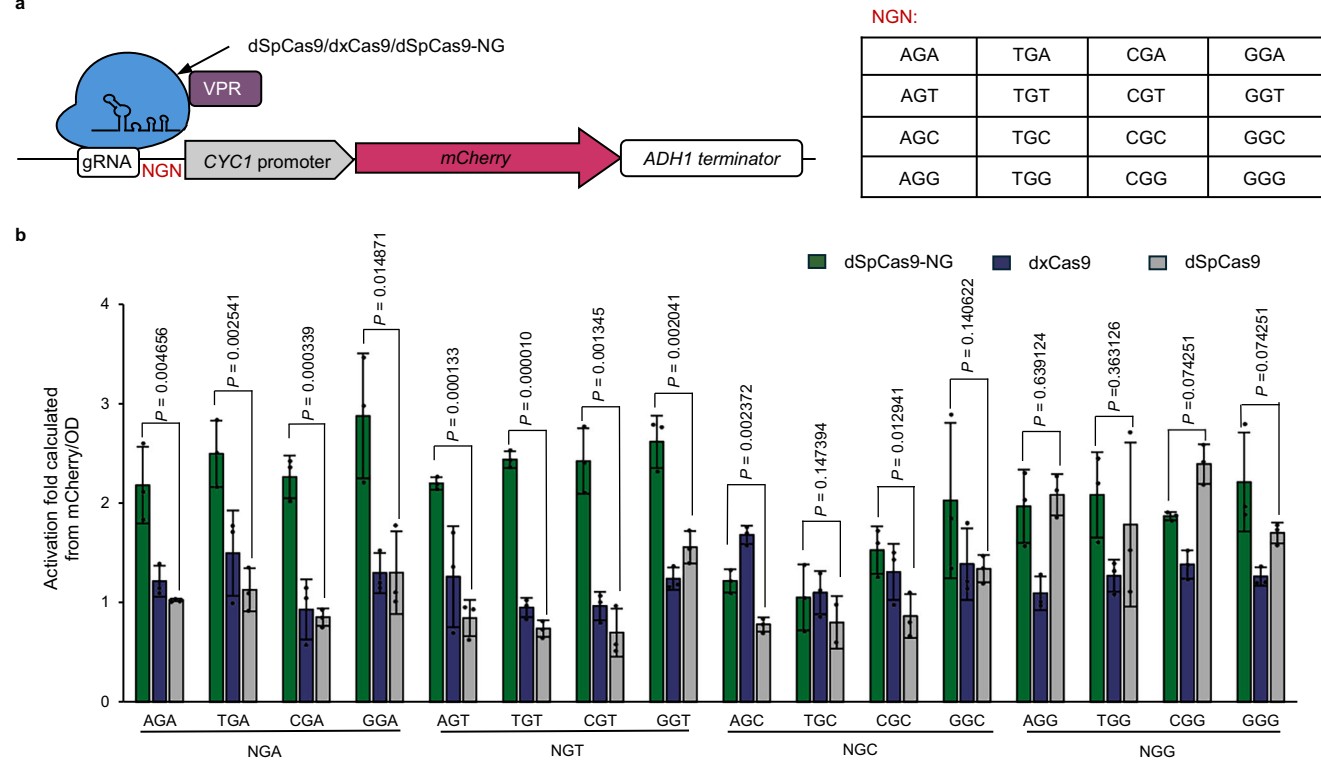

**Fig. 2 | Evaluations of dCas9 variants for activation with NG PAMs.**
**a** Construction of a yeast strain platform for dCas9 activation efficiency characterization. VPR is the abbreviation for the triple activation domain VP64-p65-Rta. **b** The characterization of each variant's activation on specific PAMs, normalized to the group containing the same variant but without gRNA. Error bars indicate mean ± SD of all replicates ($n$ = 3). Statistical analysis was performed using two-tailed Student's t-test (*$p < 0.05$, **$p < 0.01$, ***$p < 0.001$). Source data are provided as a Source Data file.

calculated as $10^7$ cells) were screened through FACS, and 4032 cells (3 rounds, twenty 96-well plates per round, with 70% screening efficiency calculated by the control plate) were selected on 96-well plates with Zeocin, and 63 mutants were validated through spot growth assays. Compared with the wild-type VPR that allows yeast tolerance up to 100 mg/L Zeocin, the final enhanced VPR mutant (e13711) could enhance yeast tolerance up to 600 mg/L Zeocin (Fig. 3f).

Finally, these mutated VPR domains were tested through real-time qPCR analysis (Fig. 3g). The result indicated that three mutants improved the expression of *mCherry* gene by 7.8-fold, 11.6-fold, and 15.5-fold, respectively. The final mutant, e13711, displayed an activation intensity almost three times that of the initial VPR, which—to the best of our knowledge—represents a record for the best CRISPR activation domains in *S. cerevisiae*. Sequences of VPR variants identified in this study can be found in Supplementary Data 3.

**Fine-tuning of the MVA pathway**
The MVA pathway is essential for eukaryotes for production of cholesterol and many terpenoids that are important for membrane integrity[33,34]. It has now been intensively used for production of a wide range of terpenoids applied in fuels, perfumes, cosmetics, as well as nutraceuticals and pharmaceuticals[35,36]. The MVA pathway in *S. cerevisiae* consists of eight genes, including *ERG10*, *ERG13*, *HMG1*, *ERG12*, *ERG8*, *ERG19*, *IDI1*, and *ERG20*, yet the ideal expression ratio of different genes has not been identified. Thus, we first applied MR to modulate the MVA pathway, using squalene production as a marker for pathway flux.

Briefly, we have introduced the mixed gRNA-tRNA array for rapid library construction, the dSpCas9-NG with expanded PAM recognition to target AT-rich promoter regions, and the mutated VPR activation

domain (e13711) with enhanced potency for combinatorial regulations. For the eight genes in the MVA pathway, we designed six gRNAs for each gene. The gRNAs of each gene were spaced by approximately 50 bp within the region between 100 bp to 500 bp upstream of the transcription start site (TSS) to avoid potential hindrance of RNA polymerase (Fig. 4a). These eight groups of gRNAs were mixed and combinatorially assembled into the p2μ-gRNA vector to create the expression library with a potential library size of $6^8$. Then, fifty colonies were randomly selected from the plate to examine the capability of MR in fine-tuning a given pathway without clear phenotype selection aids. Compared with the control strain, half of the colonies exhibited elevated squalene production ranging from 4- to 37-fold (Fig. 4b). The other half that produced less may be due to the aberrant up-regulation of gene expression ratios in the pathway, causing enzyme waste generation, intermediate accumulation, and cell stress. This result demonstrated the significant variety of our combinatorial library based on MR, as well as the importance of the method for fine-tuning unknown pathways.

The gRNAs from high producers (colony numbers 7, 14, 24, 26, and 28) were identified by sequencing and restored into the experimental strain, resulting in strains 7 R, 14 R, 24 R, 26 R, and 28 R, which maintained comparable high levels of squalene accumulation (Fig. 4c). Interestingly, the *HMG1* gRNA in these five strains was consistently gRNA 4, while the gRNAs for the other genes exhibited diversity (Fig. 4d). It has been reported that the HMG-CoA reductase, coded by *HMG1*, is one of the fluxes controlling enzymes in the MVA pathway[37–40]. Therefore, examining the strength of all *HMG1* gRNAs may help to explain the convergence of gRNA 4 in these high-producing strains. Indeed, as shown in Fig. 4e, gRNA 4 was the most effective among all designed *HMG1* gRNAs, indicating the fundamental

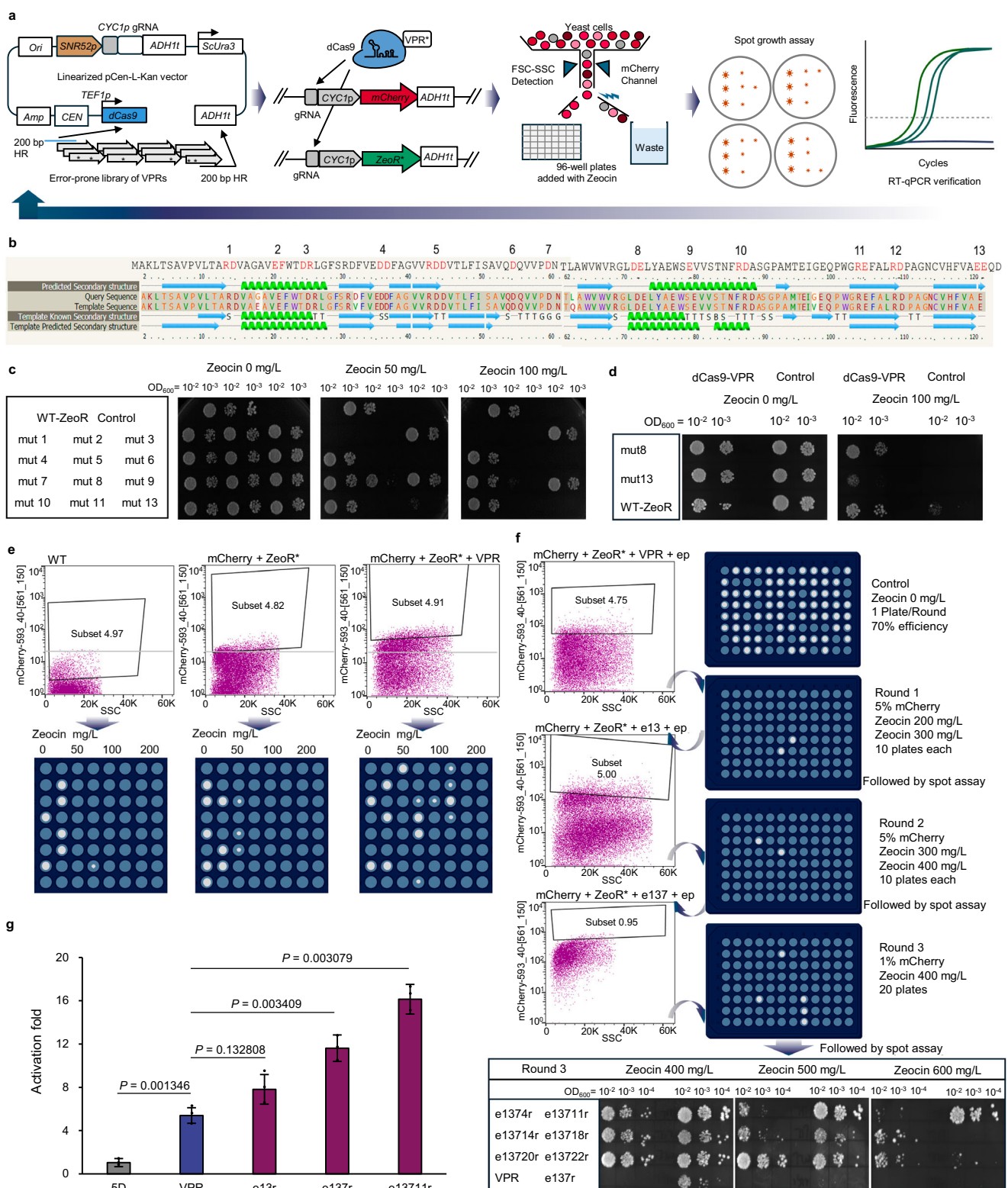

role of key gene overexpression in increasing the pathway flux, as well as the importance of the expression ratios of other genes on the final production (ranging from 12-fold to 37-fold).

Transcript changes of the five best producers, 7R, 14R, 24R, 26R, and 28R, were examined using real-time qPCR to determine whether differences in gRNA profiles altered the relative expression levels of the eight genes (Fig. 4f). The results demonstrated that despite using the same *HMG1* gRNA for the five strains, the expression of *HMG1* varied between strains, indicating an influence on *HMG1* expressions

from other genes in the pathway. Similar results have been reported that overexpression of *ERG2* and *ERG3* increased the transcription of *ERG1*[41]. As a rate-limiting gene in the MVA pathway, *HMG1* plays a crucial role in maintaining membrane structure, electron transport, protein formylation, and the biosynthesis of glycoproteins. Its expression stability is regulated by multiple transcription factors[42,43], for instance, the CGGNNNTA motif located 400 bp upstream of the TSS serves as a DNA recognition site for the zinc finger transcription factor Hap1. Nevertheless, the potential regulation of the pathway

**Fig. 3 | Enhancement of the activation capacity of CRISPR-AD VPR. a** Schematic representation of the high-throughput screening system for VPR optimization. *CEN* and HR represent the centromeric replication origin and homologous arms. VPR* denotes a mutated form of VPR. **b** Thirteen potential mutation sites to weaken Zeocin resistance. **c** Resistance evaluation of the 13 *ZeoR* gene mutants with alanine substitutions. WT-ZeoR indicates the 5D strain expressing original Zeocin resistance gene. All results show the resistance of the spots after three days, and mut12 did not grow during the seed culture stage. **d** Functional characterization of attenuated *ZeoR* gene mutants in the presence of VPR activation. **e** FACS sorting for increased *mCherry* and *ZeoR* gene expressions. WT and ZeoR* represent 5D strain and the weakened ZeoR protein. The 96-well plates were filled with SC-Ura solid medium containing a concentration gradient of Zeocin from left to right, ranging from 0, 50, 100, to 200 mg/L. The white dots represent yeast colonies grown for three days. The original images of the 96-well plates are provided in Supplementary Fig. 2. **f** Three consecutive rounds of FACS sorting followed by spot growth assay verification were conducted, with gradually increasing mCherry intensity and Zeocin concentration on plates. ep represents error-prone PCR and r indicates the reconstructing strains. Spot assay results of reconstructed VPR strains for Round 1 and 2 can be found in Supplementary Fig. 5. **g** Gene expression analysis of VPR mutants through real-time qPCR. e13r, e137r, and e13711r represent VPR mutants obtained from the first, second, and third rounds of screening, respectively. Error bars indicate mean ± SD of all replicates ($n$ = 3). Statistical analysis was performed using two-tailed Student's t-test (*$p < 0.05$, **$p < 0.01$). Source data are provided as a Source Data file.

genes would not affect the capability of MR to fine-tune the overall pathway flux, since the restored MR plasmids could generate comparable production levels as the original selected strains (Fig. 4b and c).

Moreover, although 24R and 26R had comparable *HMG1* expressions, they exhibited significant differences in squalene production, with 24R showing a 16-fold increase and 26R showing a 37-fold increase, possibly because the eight genes in 26R exhibited high expression levels, including the other generally believed key gene of the pathway, *ERG20*. Notably, the strain 28R, with both *HMG1* and *ERG20* moderately expressed, exhibited comparable squalene production to 26R, and both exhibited improved cell growth compared with the control strain (Supplementary Fig. 6), highlighting the complexity of the pathway, as well as the importance of coordinated gene expression rather than the arbitrary overexpression of the key genes.

We have then measured the concentrations of the MVA pathway metabolites in 26R and 28R to determine whether there is a potential inherent limit to the squalene production. As shown in Table 1, although the eight pathway genes were differentially regulated (Fig. 4f), the concentrations of pathway metabolites were comparable, except for MVA and MVA-PP (Table 1). Although the expression level of *ERG19* varied significantly among the three strains, all of which exhibited excessive accumulation of MVA-PP, indicating that enzymatic activity may severely limit this step. Therefore, enzyme engineering of Erg19 is expected to be an effective strategy for further optimization of the MVA pathway, thereby enhancing the performance of microbial cell factories for terpenoid production.

### Two-dimensional MR for fine-tuning the heme production pathway

Heme availability within the cell is crucial for the proper folding and function of enzymes using heme cofactors[44,45]. Due to their biological significance, heme and heme-containing proteins are key subjects in molecular cell biology, advancing fundamental research alongside applications in medicine and technology[46–50]. To demonstrate that the production improvement is not subject to overall up-regulation but rather a balanced pathway expression, we up-graded MR that could endow both up- and down-regulation. Specifically, for each of the eight genes in the heme biosynthesis pathway, six gRNAs were designed with three targeting the upstream region of the core promoter for upregulation, and the other three targeting the downstream region of the core promoter for down-regulation (Fig. 5a). Furthermore, we integrated MR with 96-well plate measurements and tested 500 colonies randomly picked from plates. With the two-dimensional combinatorial regulation, the heme production exhibited a diversity range of 152-fold (FI/OD: 5000–760,000 AU), as shown in Fig. 5b. Compared with the control strain, seven strains were identified with >10-fold increases in heme production, with a maximum increase of 17.4-fold.

To identify the potential regulatory mechanisms of the heme biosynthesis pathway, the relative expression levels of heme pathway genes were analyzed by real-time qPCR in all seven strains with >10-fold production and another three strains with <0.5-fold production.

As shown in Fig. 5c, *HEM2* exhibited completely opposite expression patterns between the high-producing and low-producing strains, with *HEM2* upregulation favoring the heme production. Specifically, according to the mRNA profile of strain 4C10, once *HEM2* was suppressed overexpressing other genes failed to improve the heme production. This result indicated *HEM2* being the potential rate-limiting gene in the pathway. Interestingly, in the best heme producing strain 4B7, all genes were moderately regulated, with each gene having strongly regulated cases in other selected strains. These results again demonstrate that balanced pathway expression outcompetes simple overexpression of all the pathway genes.

Additionally, to compare the capability of dSpCas9-NG and dSpCas9-NG on the combinatorial regulation, we built another strain library based on MR equipped with dSpCas9 and screened 500 strains as well. As expected, although the dSpCas9 system also contributed to a significant diversity of heme production, with a diversity range of 87.5-fold (FI/OD: 4000–350,000 AU) and three strains exhibiting >10-fold increases in heme production (Supplementary Fig. 7), the dSpCas9-NG system still demonstrated its strength (Fig. 5b).

## Discussion

The ideal expression of each gene endows the optimal pathway flux and reduces cell stress. Yet, this knowledge is still missing for most pathways and has to be achieved through iterative engineering. Various strategies have been used to achieve multiplexed regulations, attempting to bypass the conventional DBTL cycle. For example, CRISPR-AID[14] combines transcriptional activation, transcriptional interference, and gene deletion targeting three loci within the whole genome, and is a powerful tool identifying potential targets towards the engineering purposes. ScrABBLE[15] employed three sets of crRNA targeting three open reading frames, enabling combinatorial suppression of three genes in competing pathways. BETTER[4], GEMbLeR[5], and COMPASS[6] have also been proven to be effective combinatorial regulation strategies to fine-tune a given pathway, yet they often require major pre-integration of ribosome binding sites[4], LoxPsym sites together with upstream promoter elements and terminators[5], and transcription factor binding sites to the targeted genes[6]. Thus, an easy-to-use method with the ability to fine-tune long pathways is still needed.

In this study, we developed MR, the plug-and-tune technology that allows for fine-tuning pathways with up to eight genes in *S. cerevisiae*. This method has three key features: (1) an efficient gRNA matrix assembly system for library construction using a mixed gRNA-tRNA array; (2) an expanded PAM range, which increases targeting scope in AT-rich regions; and (3) the most potent activation domain reported in *S. cerevisiae* that provides greater regulatory amplitude. We have demonstrated the utility and efficiency of MR using both the MVA pathway and the heme biosynthesis pathway, neither of which requires high-throughput screening approaches. This is because target molecules lack biosensors or related phenotypes suitable for high-throughput screening in most scenarios, and the development of efficient combinatorial regulation methods to optimize target

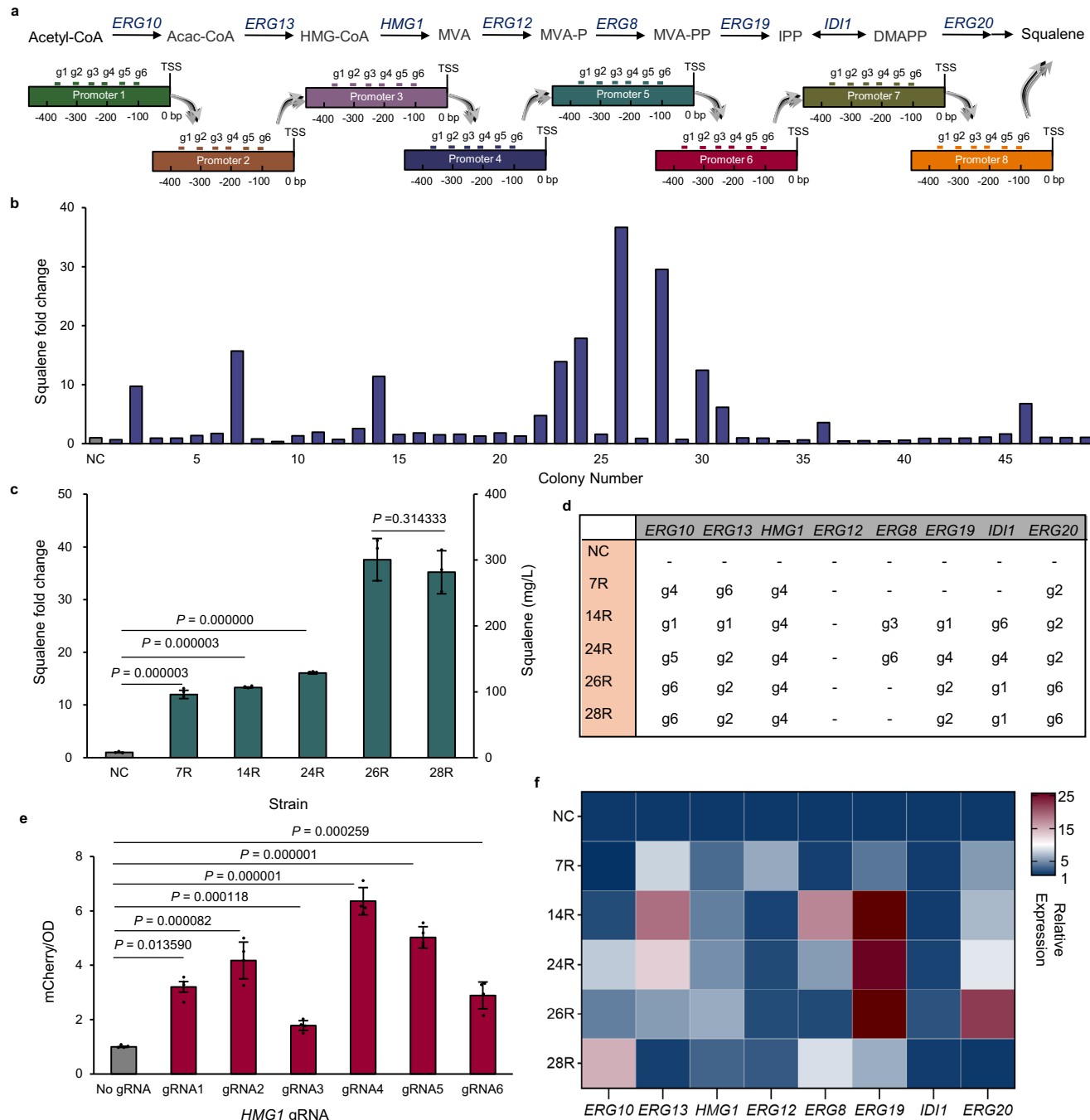

**Fig. 4 | MR enabled a broad flux diversity in the MVA pathway. a** The MVA pathway was combinatorially regulated by choosing different gRNAs. Acetyl-CoA acetyl coenzyme A, Acac-CoA acetoacetyl coenzyme A, HMG-CoA 3-hydroxy-3-methylglutaryl-CoA, MVA mevalonic acid, MVA-P mevalonic acid 5-phosphate, MVA-PP mevalonic acid 5-pyrophosphate, IPP isopentyl pyrophosphate, DMAPP dimethylallyl pyrophosphate. **b** Squalene production of the selected colonies. Fifty colonies were randomly selected from the SC-Ura-His-Leu plate. **c** Verification of the re-transformed strains. Error bars indicate mean ± SD of all replicates (*n* = 2). **d** gRNA diversity examination by Sanger sequencing. g stands for gRNA. **e** Characterization of gRNA efficiencies of *HMG1*. Error bars indicate mean ± SD of all replicates (*n* = 4). **f** Relative expression level of targeted genes. mRNA levels were analyzed by real-time qPCR experiments and normalized to *ACT1*. NC represents the strain without gRNAs. Statistical analysis was performed using two-tailed Student's *t* test (\*\*\**p* < 0.001). Source data are provided as a Source Data file.

pathways without high-throughput screening is critical. Nevertheless, integration of MR with high-throughput screening approaches and machine learning techniques will endow optimal gene expression equations of the targeted pathways, facilitating the rapid and quantitative scope of metabolic engineering.

Future optimization of MR may include: expansion of the number of target genes beyond eight by improving plasmid assembly efficiency, improvement of activators for regulatory research. Moreover,

the off-target possibilities of MR could also be determined, especially in more complex hosts. Cas9 variants have been reported with enhanced specificity[51] and targeting ability[52]. By leveraging these advanced Cas9 proteins, MR is expected to be useful in a broad number of host species.

Altogether, MR provides a more convenient and efficient option for fine-tuning pathways with multiple genes. This toolkit not only benefits fundamental research on genotype-phenotype relationships

but also accelerates metabolic engineering in yeast and sheds light on the key steps in the establishment of efficient cell factories.

## Methods

### Strains and culture conditions

The GTR-CRISPR system was used for all genome modifications in this work[13]. The *S. cerevisiae* strain CEN.PK113-5D (*MATa, MAL2-8c, SUC2, ura3-52*)[53] was used for tRNA testing and also served as the base strain for experiments in this study. To investigate the gene activation effi-

**Table 1 | Metabolite analysis of select squalene producers**

|  | Control (nM/OD$_{600}$) | 26R (nM/OD$_{600}$) | 28R (nM/OD$_{600}$) |
|---|---|---|---|
| Acetyl-CoA | 56.5 ± 5.1 | 65.7 ± 12.4 | 54.1 ± 8.3 |
| Acac-CoA | 0.8 ± 0.1 | 0.4 ± 0.0 | 0.2 ± 0.0 |
| HMG-CoA | 271.6 ± 11.4 | 130.8 ± 26.1 | 110.4 ± 4.1 |
| MVA | 4.2 ± 1.2 | 316.9 ± 68.1 | 84.5 ± 10.6 |
| MVA-P | 14.6 ± 2.9 | 21.0 ± 3.1 | 15.6 ± 0.3 |
| MVA-PP | 432.9 ± 53.1 | 303.4 ± 28.5 | 252.8 ± 15.4 |
| IPP/DMAPP | 6.3 ± 0.6 | 17.2 ± 2.6 | 12.4 ± 1.4 |
| GPP | ND | 0.11 ± 0.01 | 0.09 ± 0.02 |
| FPP | 0.19 ± 0.03 | 1.08 ± 0.25 | 1.00 ± 0.37 |
| Squalene | 8.87 ± 0.41 | 94.2 ± 4.6 | 87.8 ± 3.9 |

ND stands for not detected. Error bars represent the mean ± SD of three biological replicates. Source data are provided as a Source Data file.

*Acetyl-CoA* acetyl coenzyme A, *Acac-CoA* acetoacetyl coenzyme A, *HMG-CoA* 3-hydroxy-3-methylglutaryl-CoA, *MVA* mevalonic acid, *MVA-P* mevalonic acid 5-phosphate, *MVA-PP* mevalonic acid 5-pyrophosphate, *IPP* isopentyl pyrophosphate, *DMAPP* dimethylallyl pyrophosphate, *GPP* geranyl diphosphate, *FPP* farnesyl pyrophosphate.

ciency of different dCas9 variants on NGN PAMs without the gRNA bias, we constructed a strain platform containing 16 strains, each with *X-2::gRNA-P$_{CYC1-140bp}$-mCherry-T$_{ADH1}$* and different NGN PAM sequences. CEN.PK113-5D *X-3::P$_{CYC1-140bp}$-ZeoR-T$_{ADH1}$* was used for endogenous activator screening. CEN.PK113-5D *Gal80Δ::KanMX X-3::P$_{CYC1-140bp}$-ZeoR*-T$_{ADH1}$ XII-2::P$_{CYC1-140bp}$-mCherry-T$_{ADH1}$* was used for VPR enhancement. For the combinatorial regulation in the MVA pathway and the heme biosynthesis pathway, CEN.PK113-5D *leu2Δ his3Δ X-2::P$_{TEF1}$-dSpCas9-NG-e13711-T$_{ADH1}$* was created. For a full list of strains used in this study, see Supplementary Data 4.

For yeast transformation, a single yeast colony from a fresh plate was inoculated into YPD medium and grown overnight. The culture was then diluted into 50 mL of YPD medium to an OD$_{600}$ of 0.3. After ~5 h of growth at 30 °C, 220 rpm, when the culture reached an OD$_{600}$ of 1.6, the cells were harvested by centrifugation (3 min, 3000 rpm, 4 °C). The cell pellet was washed once with 20 mL of ice-cold 1 M sorbitol. Then the washed yeast cells were resuspended in a solution containing 16 mL of 1 M sorbitol, 2 mL of 10× TE buffer (100 mM Tris-HCl, 10 mM EDTA, pH 7.5), and 2 mL of 1 M lithium acetate. The cell suspension was incubated in a shaker for 30 min at 30 °C. Subsequently, 200 μL of 1 M dithiothreitol (DTT) was added, and incubation continued for an additional 15 min. The cells were then washed twice with 20 mL of ice-cold 1 M sorbitol. After the final wash, all supernatant was carefully removed. The resulting competent cells were resuspended in 400 μL of 1 M sorbitol. 100 μL competent cells were then dispensed into a sterile microcentrifuge tube for transformation. Less than 10 μL of the target DNAs was added to the competent cells. The mixture was subjected to electroporation at 1.5 kV using a 2 mm electroporation cuvette. After pulsing, 1 mL of 1 M sorbitol was rapidly added to the cuvette, and the cells were transferred to a 50 mL culture tube. The

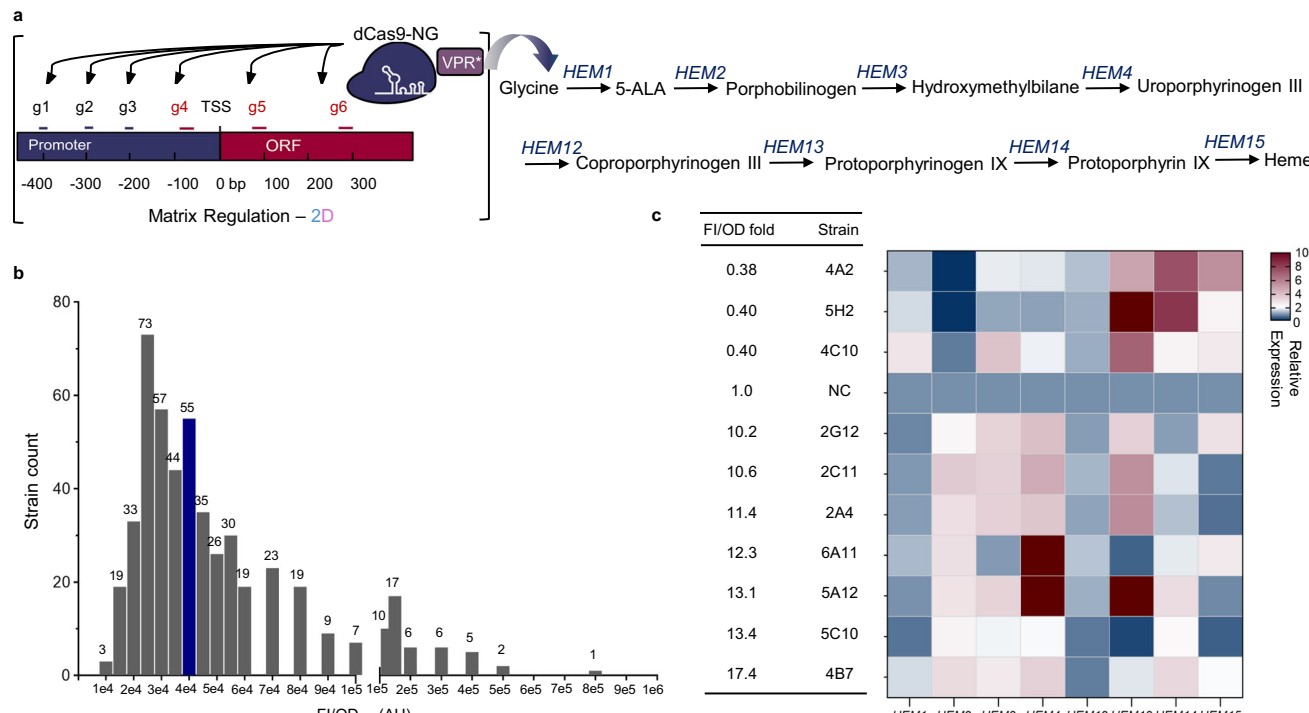

**Fig. 5 | Two-dimensional MR enabled a broad flux diversity in the heme production pathway. a** The heme biosynthesis pathway was fine-tuned by two-dimensional MR. The gRNAs (in black) designed at −200, −300, and −400 bp were used for activation, while those (in red) positioned at −100 bp, +88 bp, and +300 bp were used for repression. 5-ALA 5-aminolevulinic acid. **b** A total of 500 colonies were randomly selected and inoculated into 96-deep-well plates for heme measurement, using the strain harboring the empty plasmid as the control. Gray bars represent the distribution of test strains, while blue bar indicates the FI range of control. Seven strains with >10-fold higher production and three with <0.5-fold of the control were analyzed by real-time qPCR. **c** Relative expressions of targeted genes. mRNA levels were analyzed through real-time qPCR experiments and normalized to *ACT1*. NC represents the strain without gRNAs. Source data are provided as a Source Data file.

cuvette was washed once with an additional 1 mL of sorbitol, which was also transferred to the culture tube. Finally, 2 mL of YPD medium was added to the tube. The transformed cells were recovered by incubating the culture tube at 30 °C by shaking (250 rpm) for 5 h. Transformants were then cultured on SC agar media without the nutrient supplemented by vectors. SC-URA media supplemented with Zeocin were employed for activator screening and VPR enhancement. SC-URA-LEU-HIS media was employed to screen strains harboring plasmids for MVA pathway regulation by MR. SC-URA-LEU media were adopted for screening strains with plasmids for heme biosynthetic pathway regulation.

## Plasmid construction

A full list of plasmids used in this study can be found in Supplementary Data 4. Recombinant plasmids were constructed using Gibson Assembly or Golden Gate Assembly. The GTR-CRISPR system was used for gRNA-tRNA fragment assembly in this work[13]. pdCas9-VPR, pdxCas9-VPR, and pdSpCas9-NG-VPR for dCas9 variant characterization were constructed based on the pCas vector in the GTR-CRISPR system. The *iCas9* gene in the pCas was replaced by *dCas9-VPR*, *dxCas9-VPR*, and *dSpCas9-NG-VPR*, respectively. pCen-L-Kan vector was built by replacing the *2-micron* sequence in pCas with *CEN/ARS*, *iCas9* with *dCas9*, and *lacZα* with *KanMX*. pKan-ori-VPR vector, used as a template for the initial error-prone PCR, was created by assembling the *ori*, *KanR*, and *VPR* sequences. Subsequently, pKan-ori-e13 and pKan-ori-e137 were constructed by replacing *VPR* gene with *e13* and *e13* with *e137*, respectively. The p2μ-gRNA vector was also built on pCas by removing the *iCas9* sequence. The psgtRNA-Arg, psgtRNA-Thr, psgtRNA-Ser, and psgtRNA-Asp used for gRNA amplification were formed by replacing the *tRNA^Gly* gene sequence in psgtRNA in the GTR-CRISPR system with the relevant tRNAs. The pHis3-tRNA-Lys, pKlura3-tRNA-Leu, and pLeu2-tRNA-Ser vectors used for gRNA and marker amplification were generated by replacing *Ura3* and *tRNA^Gly* sequences in pScURA3 in the GTR-CRISPR system with the relevant markers and tRNAs.

## Library construction

In this study, all gRNA-tRNA fragments were obtained through PCR, using vectors containing different tRNAs as templates. All primers used can be found in Supplementary Data 5, and gRNA sequences in Supplementary Data 6. After gel DNA extraction, gRNAs of the same group were mixed in one tube. In 30 μL total Golden Gate mixes, 10 μL NEBridge® Ligase Master Mix buffer (M1100, New England Biolabs) and 1 μL BsaI-HFv2 (E1601, New England Biolabs) were added to the reaction mix. The quantity of vectors and fragments added was calculated using tools available at GoldenGate.neb.com. The mixtures were made according to the recommended procedure by NEBridge® Ligase Master Mix for 60 cycles. To increase the number of successful assemblies, six identical Golden Gate reactions were performed simultaneously. Subsequently, every three reactions were purified using one Cycle-pure PCR Purification Kit (Omega, USA), and finally, both columns were eluted with 35 μL of water. 10 μL of the elution was transformed into yeast strains. After a 5 h recovery, each 1 mL of the culture was plated onto the corresponding SC selective medium plates for screening.

## Extraction and quantification of squalene production

Colonies were picked from the plates and inoculated into seed cultures. After 24 h of cultivation, the cultures were inoculated to 20 mL of the selective SC medium in 100 mL shake flasks, with an initial $OD_{600}$ of 0.1. After 6 h, terbinafine was added to a final 30 mg/L concentration. Yeast cells were cultured by 72 h for squalene production determinations using a HPLC-based method at 210 nm. For squalene extraction[54], 0.2 mL of the culture was sampled and centrifuged at 3000 rpm for 5 min. Then, the cell pellet was mixed with 0.2 g acid-washed beads (0.5 mm in diameter), resuspended in 1 mL acetone, and vortexed for 20 min until residues were colorless. The acetone extracts were then centrifuged at 13,000 rpm for 10 min, filtered with a 0.45 μm nylon filter membrane before analyzing. An HPLC system (LC-20 A/SPD-20AV; SHIMADZU; LabSolutions v5.93) equipped with a C18 column (4.6 mm × 250 mm × 5 μm; SUPELCOSILTMLC-18) was used to detect the 210 nm signal at a flow rate of 1 ml/min at 40 °C. Samples were eluted with acetonitrile–methanol–isopropanol (5:3:2, v/v/v).

## Quantification of metabolites in the MVA pathway

The 26R and 28R strains carrying gRNA served as experimental samples, with the gRNA-free strain used as the negative control (NC). Cell cultures were inoculated from fresh colonies ($n = 3$) and incubated in tubes overnight and then transferred to 100 mL shake flasks with an initial $OD_{600}$ of 0.1. At mid-log phase (36 h), 1.5mL samples were collected and centrifuged at $14,000 \times g$ for 1 min, after which the cell pellet was vortexed with 250 μL of ice-cold methanol (the processed samples could be stored at −20 °C for up to one month). Prior to analysis, the samples were added with 250 μL of deionized water, vortexed, and then centrifuged at 4 °C and $14,000 \times g$ for 5 min. The supernatant was then transferred to a 3000 Da MW/CO centrifuge filter tube (Amicon Ultra from MilliporeSigma) and centrifuged at −2 °C and $13,000 \times g$ for 90 min. 400 μL of the filtrate was mixed with 500 μL of deionized water, flash-frozen in liquid nitrogen, and lyophilized for 24 h (protected from light). The pellet was then re-dissolved using 200 μL reconstitution solution (acetonitrile–methanol–water, 6:1:3, v/v/v), and measured by LC-MS.

Chromatographic separation was performed on a Waters ACQUITY UHPLC system equipped with a Waters BEH Amide column (1.7 μm, 3.0 × 100 mm) maintained at 40 °C. The injection volume was 5 μL with a constant flow rate of 400 μL/min. For positive ion mode analysis, the mobile phase consisted of (A) 10 mM ammonium formate + 0.1% ammonia in water and (B) 90% acetonitrile + 10 mM ammonium formate + 0.1% ammonia. An isocratic elution profile was employed: 30% A/70% B held from 0.00 to 5.00 min. Mass spectrometric detection was conducted using an AB Sciex 4500 triple quadrupole mass spectrometer operating in MRM mode with positive electrospray ionization. ESI source parameters were optimized as follows: ion spray voltage 4500 V, source temperature 500 °C, curtain gas 25 psi, collision gas 10 psi. Both primary and secondary mass spectrum were acquired. All data were analyzed using MultiQuant 3.0.3 software. Sample concentrations were calculated based on calibration curves constructed from the responses of reference standards versus their corresponding concentrations using the internal standard method[55].

Reagents used in this experiment: acetyl-CoA (S23156, OriLeaf), acetoacetyl-CoA (A341482, Aladdin), 3-hydroxy-3-methyl-glutaryl-CoA (H6132, Sigma), 3,5-dihydroxy-3-methylpentanoic acid (R921336, Macklin), (4S)-4-hydroxy-4-methyloxan-2-one (D869733, Macklin), mevalonic acid-5-phosphate (79849, Sigma), mevalonic acid-5-pyrophosphate (M990271, Macklin), isopentenyl diphosphate (I0503, Sigma), dimethylallyl pyrophosphate (D4287, Sigma), geranyl diphosphate (G6772, Sigma), farnesyl pyrophosphate (F6892, Sigma), DL-methionine sulfone (D863216, Macklin), deuterated d4-succinic acid (S874692, Macklin), deuterated d4-citric acid (C994619, Aladdin).

## Determination of heme concentration

For the liquid medium used in the heme experiments, 24.5 mg of ferric citrate (to achieve a final concentration of 100 μM) was first dissolved in deionized water in a Schott-Duran bottle and heated using a magnetic stirrer at 60 °C. The bottle containing the ferric citrate solution was wrapped in aluminum foil to prevent light exposure. After complete dissolution, the components of SC-URA-LEU-HIS drop-out medium and 7.507 g of glycine (to reach a final concentration of 100 mM) were added. The pH was then adjusted to 6.0, deionized water was added to a final volume of 1 L, and the medium was filter-sterilized.

Single colonies were randomly picked and inoculated into 96-well plates, with each well containing 800 μL SC-URA-LEU-HIS liquid medium supplemented with 100 mM glycine and 100 μM ferric citrate. After 12 h of cultivation, strains were subcultured into a fresh 96-well plate with started $OD_{600}$ at 0.3 and grown for an additional 24 h. The cell growth was measured. Then, 200 μL of each culture was transferred to PCR plates and centrifuged at 3000 rpm for 3 min. The pellets were resuspended in 120 μL of 20 mM oxalic acid and stored at 4 °C for 8 h. Subsequently, 120 μL of 2 M oxalic acid was added, and 120 μL of the suspension was transferred to a new PCR plate. One plate was heated at 98 °C for 30 min, while the other was incubated at room temperature for 30 min. Then, both plates were centrifuged at 3000 rpm for 3 min. 100 μL of supernatant from each plate was transferred to a black microplate for fluorescence detection using a Molecular Devices iD3 spectrophotometer (Ex: 400 nm, Em: 600 nm). The heme concentrations were calculated by subtracting the fluorescence value of the room temperature group from that of the heat-treated group.

### RNA level determinations

An exponential phase culture was used to test the RNA level using real-time qPCR with PowerUp™ SYBR™ Green Master Mix (A25742, ThermoFisher Scientific), and the total RNAs were extracted using TRIzol™ Reagent (15596026, ThermoFisher Scientific). 500 ng of the extracted RNAs were reverse transcribed into cDNA using the All-in-one™ First-Strand cDNA Synthesis Kit (QP056, GeneCopoeia). A QuantStudio 3 (ThermoFisher Scientific) instrument equipped with QuantStudio Design&Analysis Desktop Software v1.4.1 was used to collect the data and Prism 10.1.2 software was used to analyze the data. Primers for real-time qPCR in this study can be found in Supplementary Data 7.

### Assay of fluorescence intensity

Yeast colonies were inoculated to 96 deep-well plates with 800 μL of SC-URA-LEU and incubated at 30 °C, shaking at 1000 rpm for three days. Then, 50 μL of the saturated culture was diluted into 750 μL of fresh medium and cultured for one day. Fluorescence intensity was measured by a Molecular Devices iD3 plate reader spectrophotometer equipped with SoftMax Pro 7.1 software. 200 μL of the culture was transferred to a black 96-well plate for fluorescence intensity measurement, with an excitation wavelength of 578 nm and an emission wavelength of 618 nm for mCherry. Simultaneously, 10 μL of the culture was transferred to a clear-bottom microplate, and each well was diluted with 190 μL of water to measure $OD_{600}$.

### Library construction and screening of VPR mutant

The GeneMorph II Random Mutagenesis Kit (200550, Agilent, USA) was utilized to amplify the sequence for the *VPR* via error-prone PCR. Thirty PCR cycles and 500 ng template plasmid were used to obtain 3-8 mutations per gene. Purification was performed using the Cycle-pure PCR Purification Kit (Omega, USA). This fragment contained a 150 bp C-terminal sequence of dCas9 encoding sequence, an SV40 NLS, the VPR gene library, and a 198 bp ADH1 terminator. The backbone pCen-L-Kan vector was digested with the enzyme *Bsa*I at 37 °C, followed by purification. The *VPR* mutagenesis library and digested backbone dCas9 fragments were directly electroporated into yeast cells, and the pCen-L-VPR* plasmid was constructed with homology-directed repair (HDR). After a 5 h recovery, each 1 mL of yeast cells was plated onto a SC-URA plate and incubated at 30 °C for three days. The cells were then scraped off using sterile water and cultured overnight. The culture was transferred to a fresh medium and diluted to an $OD_{600}$ of 0.1, then cultured for three days in preparation for FACS. Single cells were sorted with a BD Influx (equipped with BD FACS Software sorter software) using a 561 nm laser for excitation and a 593/40 nm (RFP) filter for detection. The top 1% most fluorescent cells were collected into 96-well cell culture plates with SC-URA medium supplemented with Zeocin (R25001, Invitrogen, USA) and cultured for three days at

30 °C (Supplementary Fig. 8). The data were then analyzed using FlowJo v10.8.1 software.

### Spot growth assay

Colonies were picked from fresh SC-URA plates and inoculated into a liquid medium for overnight culture. The seed culture was then transferred to a fresh medium with an initial $OD_{600}$ of 0.1. When the culture reached an $OD_{600}$ of 0.8–1.2, the cells were diluted to $OD_{600}$ values of 0.01, 0.001, and 0.0001. Then, 5 μL of each dilution was dropped onto SC-URA plates containing Zeocin and incubated at 30 °C for 3-5 days to observe growth.

### Use of large language model

During the preparation of this work, the GPT-4 was used to ensure grammatical correction of manuscript.

### Statistics and reproducibility

The experiments were repeated at least twice with similar results to ensure reproducibility. Colonies were selected randomly from agar plates when being prepared for experimental pre-cultures; a single colony represents one biological replicate. Sample sizes were chosen based on common practices in the field and previous studies with similar experimental designs. No statistical method was used to pre-determine sample size. For tRNA characterization, the experiment was run with six biological replicates. For FI of mCherry, real-time qPCR, and growth assay, each experiment was run with three or four biological replicates. For the verification of the re-transformed strains, the experiment was run with two biological replicates. No data were excluded from the analyses. Error bars indicate mean ± SD of all replicates Statistical analysis was conducted using two-tailed unpaired *t*-tests. For all statistical analyses, $p < 0.05$ was considered significant.

### Reporting summary

Further information on research design is available in the Nature Portfolio Reporting Summary linked to this article.

## Data availability

The mass spectrometry data generated in this study for the analysis of MVA pathway metabolites are provided in Supplementary Data 8. Source data are provided with this paper.

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

## Acknowledgements

This work was supported by National Natural Science Foundation of China (22078012 and 22211530047, ZH.L.), Fundamental Research Funds for the Central Universities (buctrc202304, ZH.L.), and the Novo Nordisk Foundation (NNF10CC1016517, J.N.).

## Author contributions

X.L.T., Z.B.W., Y.P.Z., J.N., and Z.H.L. designed the research; X.L.T., Z.B.W., B.H.W., G.P.G., J.M.H., Y.F.Z., and B.Y.P. carried out the experiments; X.L.T., Z.B.W., Y.P.Z., J.Y.W., S.B.S., and Z.H.L. analysed the data; X.L.T., J.C., J.N., and Z.H.L. wrote the paper; J.N. and Z.H.L. supervised the research. X.L.T., Z.B.W., and Y.P.Z. contributed equally.

## Competing interests

The authors declare no competing interests.
