## [Transparent Peer Review file · Nature Communications]

Matrix Regulation: a plug-and-tune method for combinatorial regulation in *Saccharomyces cerevisiae*

Corresponding Author: Professor Zihe Liu

Version 0:

Reviewer comments:

Reviewer #1

(Remarks to the Author)

In this manuscript, Teng et al. describe the development of a method called Matrix Regulation (MR) that utilizes multiplex sgRNA expression, CRISPRa with SpCas9-NG, and a sgRNA library for targetable expression regulation. The authors applied the MR to endogenous mevalonate (MVA) and heme biosynthesis pathways in yeast, achieving higher titers for both.

The story presented in this manuscript is straightforward. While the results are clear, the novelty of this work is limited. The authors have combined several existing techniques, including the tRNA-sgRNA array for multiplex expression, SpCas9-NG for PAM-less CRISPR activation, and established biosynthesis pathways. The yields are expected to increase when the enzymes involved in the biosynthetic pathways are overexpressed.

As the authors note, the primary goal of fine-tuning this technology is to gain insights into the regulation of the metabolite biosynthetic pathways. Unfortunately, the authors did not provide these insights but only demonstrated that increased enzyme expression leads to higher yields. Furthermore, the increase in expression level is not stable or consistent, as indicated by the authors' own data, even though they used the same guide RNAs for the same gene.

My major concerns and explanations are:

1. Testing six other tRNAs to express multiplex sgRNAs from a tRNA-sgRNA array.

This is well known and not novel. The same laboratory reported the tRNA-sgRNA array in 2019 (Zhang et al., NC). Later, similar designs were published to express multiple sgRNAs for multiplex editing, allowing for the simultaneous targeting of up to 10 sites (Shaw et al., ACS Syn Bio 2023). Transcription of the tRNA is driven by its internal polymerase III promoters, with sgRNAs included in the same primary transcript. Following transcription, individual sgRNAs are processed and released by endogenous RNAase P and Z. This process was clearly outlined in 2019, indicating that other tRNAs can yield similar systems. In fact, several additional tRNAs have already been reported to successfully create sgRNAs, including tG(GCC)F1, tF(GAA)B, tG(GCC)E, and tG(CCC)O (Shaw et al., ACS Syn Bio 2023). Moreover, an extensive range of 13 unique tRNA genes has been evaluated for sgRNA expression to regulate dCas9 activities (including both CRISPRi and CRISPRa) in metabolic pathways within *Saccharomyces cerevisiae* (M. Deaner et al.; Biotechnology Journal, 2018). Several significant findings from this manuscript have been covered in M. Deaner et al.; Biotechnology Journal, 2018, which will be discussed further.

The author asserts that orthogonal tRNAs are necessary for library construction due to their distinct ends. However, this issue can be addressed by incorporating unique adapters for PCR onto the ends of the tRNA-Gly template. Utilizing different tRNAs can introduce additional complications, such as inconsistent expression levels and more complex construct designs, which may not be worthwhile.

Furthermore, the authors claim that the "SNR52 promoter-driven tRNA-sgRNA array becomes insufficient when the number of gRNA transcripts exceeds five." This limitation arises because the transcripts produced by the pol III SNR52 promoter tend to be short and are easily terminated by T-rich sequences. This challenge can be effectively resolved by opting for independent transcription from each tRNA instead of using the SNR52 promoter. In fact, simultaneous editing of ten targets using the tRNA-sgRNA array has been successfully achieved with greater efficiency without SNR52 (Shaw et al., ACS Syn Bio 2023).

2. dCas9-NG-VPR for gene expression regulation

The author initially combined dCas9-NG and dxCas9 for PAM-less editing but found that xCas9 was not performing

efficiently. This issue is already well-known among those working in mammalian genomic editing. The original Cas9-NG paper in *Science* in 2018 demonstrated that xCas9 struggles to recognize alternative PAMs efficiently. Several improved versions were developed, such as xCas9-NG, but were not mentioned in this manuscript. Furthermore, the activation of gene expression using dCas9-VPR with the PAM sequence NGG has been reported numerous times.

The authors then improved gene activation by testing various VPR mutants. They successfully isolated a mutant that exhibited three times the activation level compared to the original VPR. This is a useful observation.

Finally, the authors combined Cas9-NG with their improved VPR domain for PAM-less CRISPR activation (CRISPRa).

While their approach allows for a broader selection of guide RNAs (gRNAs), it presents a significant challenge: Cas9-NG exhibits markedly lower efficiency than the original Cas9-NGG. As a result, researchers working with yeast continue to prefer Cas9-NGG, which typically achieves editing efficiencies close to 100%. In contrast, Cas9-NG shows considerably lower efficacy, often rendering it ineffective—even when using the same functional gRNA as with the NGG PAM. This behavior is different from the observations in mammalian systems. It has been speculated—though further investigation is needed—that Cas9-NG has a considerably lower binding affinity than Cas9-NGG. Consequently, dCas9-NG-VPR may not be the optimal choice for CRISPRa in yeast. In summary, while dCas9-NG offers access to a more diverse set of gRNAs, it comes with significantly reduced efficiency for CRISPR activation.

Therefore, an essential control experiment that the authors need to include is the use of the same gRNA with an NGG PAM to demonstrate their Cas9-NG system can achieve comparable activation efficiency to Cas9-NGG. If this is not the case, then dCas9-NGG should be favored over dCas9-NG for CRISPRa applications.

3. The author constructed a CRISPRa library for inducible gene expression in the endogenous MVA and heme biosynthesis pathways as a demo.

Firstly, CRISPR activation for the expression of endogenous pathways has been previously established, including in the referenced paper (M. Deaner et al.; *Biotechnology Journal*, 2018). The multiplex metabolic pathway engineering of the same five-gene MVA pathway was also reported in 2015 (<https://doi.org/10.1016/j.ymben.2015.01.008>). Please revise the statement in Line 56. Both pathways have been established previously. Regarding the heme pathway, why were HEM1 and HEM4 not included?

Therefore, the methodology and targets for pathway engineering are not novel, neither do the authors provide insights into the regulation of the studied pathways. While there is an increase in the yield of the final products, it remains unclear whether this is due to a general upregulation of all pathway enzymes or the result of precise fine-tuning within the respective pathway. The observed inconsistency is a common issue associated with CRISPRa, as achieving stable increases in target gene expression to a specific level can be challenging. The data presented by the author also reveal inconsistent expression levels despite using the same gRNA for identical targets across five strains as biological replicates. This variability significantly limits the application of this approach in metabolic engineering, undermining the author's assertion of yield enhancement, which may only reflect an occasional outcome.

Moreover, this method still necessitates labor-intensive screening to identify the optimal titers of the metabolite of interest, and due to the random selection of clones, there is a high chance of missing out on the best combination of gene regulation.

Reviewer #2

(Remarks to the Author)

This manuscript describes the development of Matrix Regulation (MR), a CRISPR-mediated pathway fine-tuning method that enables the construction of 68 combinations and screening for the optimal transcriptional expression of pathways with up to eight genes in *Saccharomyces cerevisiae*. Key aspects of MR include the identification and characterization of multiple tRNAs to facilitate gRNA splicing and matrix assembly, the selection of dCas9 variants, particularly dSpCas9-NG, which broadens the targeting scope from NGG PAM to NG PAM sequences, and the screening and enhancement of ADs, that significantly increases the dynamic range of CRISPR-based activation in yeast. The approach was demonstrated to be effective in both the mevalonate pathway, leading to a 37-fold increase in squalene production and a 10-fold increase in heme production. The article is well written, with clear experiments, and its contents will be of interests to the readership of synthetic biology and metabolic engineering. Some minor questions are needed before publication.

1. In Line 106, what were the criteria for selecting the genes CAN1, LYP1, TRP2, FAA1, and ADE2?

2. In Line 135-137, the statement needs to be strengthened with citations.

3. In Line 139-141, with so many proteins in the *Saccharomyces* Genome Database, how to identify these 101 candidates? How to cover all genes with regulatory capability?

4. Have researchers conducted similar mutation studies? If not, incorporating the procedure into the main text would make it clearer for readers on how to proceed.

5. All the figures are lacking statistics, please add.

6. In the fine-tuning the heme production pathway, do the strains with increased heme production share any common characteristics? Is it possible to analyze the relevant genes in the same way as analyzed in the fine-tuning the MVA pathway mentioned above? Since HMG1 in the MVA pathway has been clearly elucidated by researchers, and heme production being a hot topic, the correlation of these genes is likely to be of greater interest to researchers.

7. It is suggested to strengthen the Discussion part. The current discussion section merely summarizes the content of the main text, which has already been mentioned above. I suggest adding some discussion here that compares this MR with existing research.

Reviewer #3

(Remarks to the Author)

Matrix Regulation: a plug-and-tune method for combinatorial regulation in *Saccharomyces cerevisiae*
Xiaolong Teng, Zibai Wang, Yueping Zhang, Binhao Wang, Guiping Gong, Jinmiao Hu, Yifan Zhu, Baoyi Peng, James Chen, Shuobo Shi, Jens Nielsen, and Zihe Liu

The manuscript reports “Matrix Regulation” (MR), a CRISPR mediated method to enable optimization of transcriptional regulation in pathways with up to eight genes. The authors had to develop several tools to make this workable namely (i) creating a mixed gRNA-tRNA array for rapid assembly of a gRNA matrix for library construction, (ii) the “dSpCas9-NG” variant Cas nuclease with expanded PAM recognition to target AT-rich promoter regions and (iii) the mutated VPR activation domain (e13711) with enhanced potency for combinatorial matrix regulations. These were then applied to generate combinatorial libraries of the mevalonate pathway (8 gene promoters) and the heme biosynthesis pathway (6 gene promoters), respectively, and increased squalene and heme production by 37-fold and 10-fold, respectively. This demonstrates the versatility of the approach and its applicability in fundamental and biotechnological research. The manuscript is well-written and presents novel, impactful data, warranting recommendation for publication once the relatively minor comments below have been addressed.

Comments:

1. Have the authors considered the possibility of off-target effects? CRISPR-Cas systems are known to occasionally introduce off-target modifications, which could potentially be detected using RNA-Seq (though this is not necessary for this study). The fact that the growth of the squalene-optimized strains was not negatively affected (Sup. Fig. 6) suggests that off-target effects were likely not a significant issue. However, the authors should discuss how off-target effects might influence the reliability and applicability of this method.
2. In lines 139 – 141 the authors state: “We started by identifying potent endogenous activators from 101 candidates through the *Saccharomyces* Genome Database (SGD). Specifically, all genes with regulatory capability descriptions were included.” However, the supplementary material does not provide references to where the regulatory descriptions of the genes were provided. This should be added.
3. In Figure 3, the growth/no growth phenotype in the image is quite unclear. Can a heat map be substituted with the images moved to supplementary material?
4. Figure 3d: statistically significant differences should be indicated on this bar chart to support the conclusions made.
5. Line 179: the term “1 mL culture with 5 OD600” is unclear. Similarly, line 356: “amount of 8 OD600 of cells were collected” Please clarify.
6. Regarding figure 4f: Lines 228 to 230 authors state: “despite using the same HMG1 gRNA for the five strains, the expression of HMG1 varied between strains, indicating an influence on HMG1 expressions from other genes.” How would the expression of the other genes influence the expression of the HMG1 in these strains?
7. Fig. 4f: it is notable that the expression levels of the mevalonate pathway genes in 28R and 26R differed for the two strains, yet the squalene produced was statistically similar (Fig. 4C). Does this convergence indicate a possible inherent limit to squalene production? The authors only mention that this points to the importance of coordinated gene expression as opposed to arbitrary overexpression. Metabolic flux analysis may provide further clues as to the optimal levels of the pathway (though again this may be beyond the current scope). However, if the inherent limit of squalene production has not yet been reached, this may indicate that pathway regulation is still suboptimal. The authors demonstrated that two different “routes” achieved the same “destination,” but it is possible that neither is fully optimized. It would be valuable for the authors to comment on whether screening a larger number of colonies could reveal even more effective combinations.

Minor comments:

1. Line 92: Replace “knockouts employs a” with “knockouts which employs a”
2. Line 101: Replace “with distinct four nucleotides at the 3’ end, including” with “with four distinct nucleotides at the 3’ end, namely”
3. Line 134: Replace “use of protein” with “use of a protein”
4. Line 136: Replace “use” with “used”
5. Line 149: Replace “5% genes” with “5% of the genes”
6. Line 211: Replace “mis-up-regulation” with “aberrant up-regulation”
7. Line 256: Replace “ these knowledges for most pathways are still missing” with “this knowledge is still missing for most pathways”
8. Line 308: add a reference for the transformation method.
9. Line 369: Replace “revers” with “reverse”
10. Line 478: Replace “on” with “no”

Version 1:

Reviewer comments:

Reviewer #2

(Remarks to the Author)

I believe the authors have answered my questions well and the manuscript is acceptable.

Reviewer #3

(Remarks to the Author)

The authors have satisfactorily addressed my previous concerns. I would recommend publication of the manuscript.

Reviewer 1:

In this manuscript, Teng et al. describe the development of a method called Matrix Regulation (MR) that utilizes multiplex sgRNA expression, CRISPRa with SpCas9-NG, and a sgRNA library for targetable expression regulation. The authors applied the MR to endogenous mevalonate (MVA) and heme biosynthesis pathways in yeast, achieving higher titers for both.

The story presented in this manuscript is straightforward. While the results are clear, the novelty of this work is limited. The authors have combined several existing techniques, including the tRNA-sgRNA array for multiplex expression, SpCas9-NG for PAM-less CRISPR activation, and established biosynthesis pathways. The yields are expected to increase when the enzymes involved in the biosynthetic pathways are overexpressed.

As the authors note, the primary goal of fine-tuning this technology is to gain insights into the regulation of the metabolite biosynthetic pathways. Unfortunately, the authors did not provide these insights but only demonstrated that increased enzyme expression leads to higher yields. Furthermore, the increase in expression level is not stable or consistent, as indicated by the authors' own data, even though they used the same guide RNAs for the same gene.

We thank the reviewer for the constructive comments, as these have enabled us to improve the quality of the manuscript. Accordingly, we have now carefully revised the manuscript with more experiments and more discussions. For example, we have now added a two-dimensional matrix regulation of the heme biosynthesis pathway, with each of the pathway genes up-regulated at three levels and down-regulated at three levels. The results showed that the best producers were not the strains with all pathway genes overexpressed, which allowed us to identify new regulations. We feel confident that all issues have been addressed and hope that the reviewer will find the revised manuscript improved.

Major concerns:

1. Testing six other tRNAs to express multiplex sgRNAs from a tRNA-sgRNA array.

This is well known and not novel. The same laboratory reported the tRNA-sgRNA array in 2019 (Zhang et al., NC). Later, similar designs were published to express multiple sgRNAs for multiplex editing, allowing for the simultaneous targeting of up to 10 sites (Shaw et al., ACS Syn Bio 2023). Transcription of the tRNA is driven by its internal

polymerase III promoters, with sgRNAs included in the same primary transcript. Following transcription, individual sgRNAs are processed and released by endogenous RNAase P and Z. This process was clearly outlined in 2019, indicating that other tRNAs can yield similar systems. In fact, several additional tRNAs have already been reported to successfully create sgRNAs, including tG(GCC)F1, tF(GAA)B, tG(GCC)E, and tG(CCC)O (Shaw et al., ACS Syn Bio 2023). Moreover, an extensive range of 13 unique tRNA genes has been evaluated for sgRNA expression to regulate dCas9 activities (including both CRISPRi and CRISPRa) in metabolic pathways within *Saccharomyces cerevisiae* (M. Deaner et al.; Biotechnology Journal, 2018). Several significant findings from this manuscript have been covered in M. Deaner et al.; Biotechnology Journal, 2018, which will be discussed further.

The author asserts that orthogonal tRNAs are necessary for library construction due to their distinct ends. However, this issue can be addressed by incorporating unique adapters for PCR onto the ends of the tRNA-Gly template. Utilizing different tRNAs can introduce additional complications, such as inconsistent expression levels and more complex construct designs, which may not be worthwhile.

Furthermore, the authors claim that the "SNR52 promoter-driven tRNA-sgRNA array becomes insufficient when the number of gRNA transcripts exceeds five." This limitation arises because the transcripts produced by the pol III SNR52 promoter tend to be short and are easily terminated by T-rich sequences. This challenge can be effectively resolved by opting for independent transcription from each tRNA instead of using the SNR52 promoter. In fact, simultaneous editing of ten targets using the tRNA-sgRNA array has been successfully achieved with greater efficiency without SNR52 (Shaw et al., ACS Syn Bio 2023)

We thank the reviewer for the insightful queries and suggestions, and we are so sorry that we did not demonstrate the rationale of our design clearly, which caused the confusion.

Indeed, the tRNAs we evaluated were not all new for CRISPR-based regulation, but rather more suitable for the library construction of the Matrix Regulation in yeast. We have now modified this part in lines 104-111 to make it clearer.

In principle, introducing unique adapters for distinct ends while reserving tRNA-Gly for all target genes could also work for the Matrix Regulation design. Yet, this may cause more possibilities of mis-constructed plasmids due to the highly efficient

homologous recombination efficiency of yeast. Briefly, for each gene to be regulated, the same 76-bp scaffold RNA has to be assembled, which together with 71 bp of tRNA-Gly would be 147 bp. Whereas in *S. cerevisiae* 100-bp homology arms would cause efficient gap repairs, especially in Matrix Regulation where all components are directly transformed into yeast rather than pre-assembled in vitro. For example, our 2019 study on the tRNA-sgRNA arrays (Zhang et al., NC) demonstrated that the tRNA-Gly-only pattern failed to achieve simultaneous eight-gene disruption, and we suggested that it “could be due to a loop-out issue or low plasmid construction efficiencies with large number of repetitive sequences.” These discussions have now been added to the revised manuscript.

Indeed, "*SNR52* promoter-driven tRNA-sgRNA array becomes insufficient when the number of gRNA transcripts exceeds five" can be effectively resolved by opting for independent transcription from each tRNA instead of using the *SNR52* promoter. We used this format only for the screening of alternative tRNAs that are comparable with tRNA-Gly in our application. We have now clarified this part in line 115-121 to make it clearer.

2. dCas9-NG-VPR for gene expression regulation.

The author initially combined dCas9-NG and dxCas9 for PAM-less editing but found that xCas9 was not performing efficiently. This issue is already well-known among those working in mammalian genomic editing. The original Cas9-NG paper in Science in 2018 demonstrated that xCas9 struggles to recognize alternative PAMs efficiently. Several improved versions were developed, such as xCas9-NG, but were not mentioned in this manuscript. Furthermore, the activation of gene expression using dCas9-VPR with the PAM sequence NGG has been reported numerous times.

The authors then improved gene activation by testing various VPR mutants. They successfully isolated a mutant that exhibited three times the activation level compared to the original VPR. This is a useful observation.

Finally, the authors combined Cas9-NG with their improved VPR domain for PAM-less CRISPR activation (CRISPRa). While their approach allows for a broader selection of guide RNAs (gRNAs), it presents a significant challenge: Cas9-NG exhibits markedly lower efficiency than the original Cas9-NGG. As a result, researchers working with yeast continue to prefer Cas9-NGG, which typically achieves editing efficiencies close to 100%. In contrast, Cas9-NG shows considerably lower efficacy,

often rendering it ineffective—even when using the same functional gRNA as with the NGG PAM. This behavior is different from the observations in mammalian systems. It has been speculated—though further investigation is needed—that Cas9-NG has a considerably lower binding affinity than Cas9-NGG. Consequently, dCas9-NG-VPR may not be the optimal choice for CRISPRa in yeast. In summary, while dCas9-NG offers access to a more diverse set of gRNAs, it comes with significantly reduced efficiency for CRISPR activation.

Therefore, an essential control experiment that the authors need to include is the use of the same gRNA with an NGG PAM to demonstrate their Cas9-NG system can achieve comparable activation efficiency to Cas9-NGG. If this is not the case, then dCas9-NGG should be favored over dCas9-NG for CRISPRa applications.

We thank the reviewer for the insightful queries and suggestions. Indeed, while only Cas9-NG and xCas9 with NG PAM specificity had been reported when we initiated this project, through the years other versions have been identified. Yet, in our previous work (Gong et al., ACS Syn Bio 2021) we demonstrated that Cas9-NG could generate highly efficient genome disruptions approaching 100% for single-gene knockouts across all NGN PAMs. These results indicate that Cas9-NG should be efficient enough for genome targeting, we have thus stick to Cas9-NG for the following study. The dCas9 system with NGG PAM was employed solely as a control for Cas9-NG and xCas9 for their activation activities. We have now clarified these points in lines 131-135 of the revised manuscript.

Indeed, it is necessary to compare Cas9-NG with Cas9-NGG using the same system and the same gRNA. As shown in Fig. 2b, using the same gRNAs Cas9-NG exhibited comparable activation efficiencies to Cas9-NGG when targeting the NGG PAM sequences (p-values ranging from 0.07 to 0.36) and exhibited significantly higher activation efficiencies to Cas9-NGG for NGA, NGT, and NGC PAM sequences. Moreover, we have now added Matrix Regulation experiments using both Cas9-NG and Cas9-NGG for heme production, which demonstrated that Cas9-NG exhibited better regulation capacities, presenting more heme overproducing strains and improved production levels. We have now clarified these points in lines 314-319, Fig. 5b, and Supplementary Fig. 7 of the revised manuscript.

3. The author constructed a CRISPRa library for inducible gene expression in the endogenous MVA and heme biosynthesis pathways as a demo.

Firstly, CRISPR activation for the expression of endogenous pathways has been previously established, including in the referenced paper (M. Deaner et al.; Biotechnology Journal, 2018). The multiplex metabolic pathway engineering of the same five-gene MVA pathway was also reported in 2015 (<https://doi.org/10.1016/j.ymben.2015.01.008>). Please revise the statement in Line 56. Both pathways have been established previously. Regarding the heme pathway, why were *HEM1* and *HEM4* not included?

Therefore, the methodology and targets for pathway engineering are not novel, neither do the authors provide insights into the regulation of the studied pathways. While there is an increase in the yield of the final products, it remains unclear whether this is due to a general upregulation of all pathway enzymes or the result of precise fine-tuning within the respective pathway. The observed inconsistency is a common issue associated with CRISPRa, as achieving stable increases in target gene expression to a specific level can be challenging. The data presented by the author also reveal inconsistent expression levels despite using the same gRNA for identical targets across five strains as biological replicates. This variability significantly limits the application of this approach in metabolic engineering, undermining the author's assertion of yield enhancement, which may only reflect an occasional outcome.

Moreover, this method still necessitates labor-intensive screening to identify the optimal titers of the metabolite of interest, and due to the random selection of clones, there is a high chance of missing out on the best combination of gene regulation.

We thank the reviewer for the very constructive suggestions. We have now revised the statement in line 56 (now in lines 57-63) focusing on the rationale for developing CRISPR systems with both expanded targeting scope and enhanced activation domains for combinatorial pathway regulation. We have also performed additional experiments implementing screening of the heme pathway including all genes, which provided us with new insights into the pathway. We hope the reviewer will find this part improved (lines 286-319 of the revised manuscript).

Briefly, we have now up-graded Matrix Regulation to two-dimensional combinatorial regulations (i.e., with both up- and down-regulation of the pathway genes in varied degrees) and used it to regulate all the eight genes in the heme biosynthetic pathway. We have then used both Cas9-NG and Cas9-NGG derived Matrix Regulation to screen 500 strains for heme productions, respectively. The best heme producing strains we selected exhibited 17.4-fold increase compared with that of the control strain, yet with

relatively moderate regulated pathway genes and even with *HEM12* slightly down-regulated, demonstrating that balanced expression profiles are more beneficial for targeted overproductions.

Indeed, we acknowledge that although sharing the same gRNA the expressions *HMG1* of in the selected strains varied by ~2 fold (between 3-fold to 6.2-fold than that of the control strain). Similar results have been reported that when the expression of other pathway genes changed, the transcription of unchallenged gene also changed. For example, it was reported that the overexpression of *ERG2* and *ERG3* also caused up-regulation of *ERG1* (Front Microbiol 2022, DOI: [10.3389/fmicb.2022.978074](https://doi.org/10.3389/fmicb.2022.978074)). We speculated that this may be caused by some feed-back regulatory machineries for *HMG1*. For example, the CGGNNNTA motif located 400 bp upstream of the TSS of *HMG1* could serve as a DNA recognition site for the zinc finger transcription factor Hap1. Nevertheless, the combinatorial regulation could still generate stable regulation to the pathway. For example, the gRNA plasmids extracted from high-yield strains were transformed into a clean background strain and could generate comparable squalene over-productions. We have now added in-depth discussion in the revised manuscript (lines 259-268).

We agree that without high-throughput screening it could be labor-intensive to identify the optimal titers of the metabolite of interest. The primary goal of this manuscript is to present an easy-to-use method with the ability to fine-tune long pathways. We intended to use random selection of clones for both cases to demonstrate the strength of the method, which was capable to improve production by 37-fold and 17-fold, respectively. We foresee that when equipped with high-throughput screening, the workflow could be substantially simplified.

Reviewer 2:

This manuscript describes the development of Matrix Regulation (MR), a CRISPR-mediated pathway fine-tuning method that enables the construction of 6^8 combinations and screening for the optimal transcriptional expression of pathways with up to eight genes in *Saccharomyces cerevisiae*. Key aspects of MR include the identification and characterization of multiple tRNAs to facilitate gRNA splicing and matrix assembly, the selection of dCas9 variants, particularly dSpCas9-NG, which broadens the targeting scope from NGG PAM to NG PAM sequences, and the screening and enhancement of ADs, that significantly increases the dynamic range of CRISPR-based activation in yeast. The approach was demonstrated to be effective in both the mevalonate pathway, leading to a 37-fold increase in squalene production and a 10-fold increase in heme production. The article is well written, with clear experiments, and its contents will be of interests to the readership of synthetic biology and metabolic engineering. Some minor questions are needed before publication.

We thank the reviewer for the kind and constructive comments, as these have enabled us to improve the quality of the paper. We have now revised the manuscript based on all the comments. We hope the reviewer would find our manuscript improved.

1. In Line 106, what were the criteria for selecting the genes *CAN1*, *LYP1*, *TRP2*, *FAA1*, and *ADE2*?

We thank the reviewer for pointing out the logical gap in target selection for characterizing the slicing capability of candidate tRNAs. The gRNAs of these genes are well characterized and used in several papers including our previous paper demonstrating the gRNA-tRNA array. To avoid the gRNA efficiency bias and to get a quick comparison of alternative tRNAs, we thus stick to the five genes. We have now discussed this rationality in the revised manuscript (lines 115-121).

2. In Line 135-137, the statement needs to be strengthened with citations.

We thank the reviewer for this suggestion, and we have now added the citation in the revised manuscript. Briefly, studies reported that multiple effectors, even in the absence of crRNA, could reduce the cell growth (Zhai et al., *Nucleic Acids Res* 2022; Barajas et al., *Nature Commun* 2022).

3. In Line 139-141, with so many proteins in the *Saccharomyces* Genome Database, how to identify these 101 candidates? How to cover all genes with regulatory capability?

We thank the review for pointing out the missing details for selecting the 101 candidates from the SGD. Briefly, we identified the potential candidates following three criteria: (1) Annotated as transcription factors, activators, or regulators; (2) With no reported functions of repression or inhibition; (3) For those containing DNA-binding domains, the remaining parts were investigated separately. We have now added discussions in lines 157-162 in the revised manuscript.

4. Have researchers conducted similar mutation studies? If not, incorporating the procedure into the main text would make it clearer for readers on how to proceed.

We thank the reviewer for pointing this out. We have now incorporated the procedure into the Results section (lines 185-189 and Fig. 3b-d).

5. All the figures are lacking statistics, please add.

We are so sorry for missing statistical data, which have now been added in the revised manuscript.

6. In the fine-tuning the heme production pathway, do the strains with increased heme production share any common characteristics? Is it possible to analyze the relevant genes in the same way as analyzed in the fine-tuning the MVA pathway mentioned above? Since *HMG1* in the MVA pathway has been clearly elucidated by researchers, and heme production being a hot topic, the correlation of these genes is likely to be of greater interest to researchers.

We thank the reviewer for the constructive suggestions to gain deeper insights into the heme pathway. We have now up-graded Matrix Regulation to two-dimensional combinatorial regulations (i.e., both up- and down-regulation of the pathway genes in varied degrees) and used it to regulate all the eight genes in the heme biosynthetic pathway. We have then screened 500 strains through 96-well plates, among which seven strains were identified with > 10-fold increase in heme productions (with a maximum of 17.4-fold). Further characterization of these strains identified *HEM2* as the bottleneck gene strictly controlling flux through the heme pathway and the overexpression of the downstream *HEM14* gene was established as another critical factor to enhance heme production. Importantly, our findings challenge the conventional approach of overall or arbitrary overexpression of pathway genes, demonstrating that balanced expression profiles are more beneficial to both strain optimization and industrial scale-up in yeast cell factories. We hope the reviewer will find this part improved (lines 286-313 of the revised manuscript).

7. It is suggested to strengthen the Discussion part. The current discussion section merely summarizes the content of the main text, which has already been mentioned above. I suggest adding some discussion here that compares this MR with existing research.

We thank you for this valuable suggestion. We have now strengthened the Discussion part comparing Matrix Regulation with existing CRISPR-based and non-CRISPR-based strategies for combinatorial regulations, as shown in lines 329-338 in the revised manuscript.

Reviewer 3:

The manuscript reports “Matrix Regulation” (MR), a CRISPR mediated method to enable optimization of transcriptional regulation in pathways with up to eight genes. The authors had to develop several tools to make this workable namely (i) creating a mixed gRNA-tRNA array for rapid assembly of a gRNA matrix for library construction, (ii) the “dSpCas9-NG” variant Cas nuclease with expanded PAM recognition to target AT-rich promoter regions and (iii) the mutated VPR activation domain (e13711) with enhanced potency for combinatorial matrix regulations. These were then applied to generate combinatorial libraries of the mevalonate pathway (8 gene promoters) and the heme biosynthesis pathway (6 gene promoters), respectively, and increased squalene and heme production by 37-fold and 10-fold, respectively. This demonstrates the versatility of the approach and its applicability in fundamental and biotechnological research.

The manuscript is well-written and presents novel, impactful data, warranting recommendation for publication once the relatively minor comments below have been addressed.

We thank the reviewer for the kind and constructive suggestions. We have now carefully revised the manuscript accordingly, and we hope the reviewer would find our manuscript improved.

1. Have the authors considered the possibility of off-target effects? CRISPR-Cas systems are known to occasionally introduce off-target modifications, which could potentially be detected using RNA-Seq (though this is not necessary for this study). The fact that the growth of the squalene-optimized strains was not negatively affected (Sup. Fig. 6) suggests that off-target effects were likely not a significant issue. However, the authors should discuss how off-target effects might influence the reliability and applicability of this method.

We thank the reviewer for the kind suggestion regarding the impact of off-target effects on Matrix Regulation. In the revised manuscript in lines 356-359, we have now included an in-depth discussion on the influence of off-target effects and potential control strategies.

2. In lines 139 – 141 the authors state: “We started by identifying potent endogenous activators from 101 candidates through the Saccharomyces Genome Database (SGD).

Specifically, all genes with regulatory capability descriptions were included.” However, the supplementary material does not provide references to where the regulatory descriptions of the genes were provided. This should be added.

We are so sorry for the missing references, which have now been added accordingly.

3. In Figure 3, the growth/no growth phenotype in the image is quite unclear. Can a heat map be substituted with the images moved to supplementary material?

We thank the reviewer for the suggestions regarding the adjustments to Fig. 3e-f. In the revised manuscript, we have now replaced the figures with growth phenotypes with heat map figures accordingly and moved the original figures to Supplementary Fig. 2.

4. Figure 3d: statistically significant differences should be indicated on this bar chart to support the conclusions made.

We are so sorry for the missing statistical data. We have now added the information in the revised manuscript.

5. Line 179: the term “1 mL culture with 5 OD₆₀₀” is unclear. Similarly, line 356: “amount of 8 OD₆₀₀ of cells were collected” Please clarify.

We are so sorry for the ambiguity in the expressions. We have modified them into “three rounds of FACS sorting were performed, each using 1 mL of cell culture at an OD₆₀₀ of 5”, and “After 12 hours of cultivation, cultures were subcultured into a fresh 96-well plate with started OD₆₀₀ at 0.3 and grown for an additional 24 hours.”, respectively.

6. Regarding figure 4f: Lines 228 to 230 authors state: “despite using the same *HMGI* gRNA for the five strains, the expression of *HMGI* varied between strains, indicating an influence on *HMGI* expressions from other genes.” How would the expression of the other genes influence the expression of the *HMGI* in these strains?

We feel sorry to have caused the confusion. The expressions *HMGI* of in the selected strains sharing the same gRNA varied by ~2 fold (between 3-fold to 6.2-fold than that of the control strain). Similar results have been reported that when the expression of other pathway genes changed, the transcription of unchallenged gene also changed. For example, it was reported that the overexpression of *ERG2* and *ERG3* also caused up-regulation of *ERGI* (Front Microbiol 2022, DOI: [10.3389/fmicb.2022.978074](https://doi.org/10.3389/fmicb.2022.978074)). Since

HMG1 has been suggested to be one of the rate-limiting enzymes of the mevalonate pathway, we speculated that the strain may possess some feed-back regulatory machineries for *HMG1*. For example, the CGGNNNTA motif located 400 bp upstream of the TSS of *HMG1* could serve as a DNA recognition site for the zinc finger transcription factor Hap1. We have now added an in-depth discussion of this observation (lines 259-268).

7. Fig. 4f: it is notable that the expression levels of the mevalonate pathway genes in 28R and 26R differed for the two strains, yet the squalene produced was statistically similar (Fig. 4C). Does this convergence indicate a possible inherent limit to squalene production? The authors only mention that this points to the importance of coordinated gene expression as opposed to arbitrary overexpression. Metabolic flux analysis may provide further clues as to the optimal levels of the pathway (though again this may be beyond the current scope). However, if the inherent limit of squalene production has not yet been reached, this may indicate that pathway regulation is still suboptimal. The authors demonstrated that two different "routes" achieved the same "destination," but it is possible that neither is fully optimized. It would be valuable for the authors to comment on whether screening a larger number of colonies could reveal even more effective combinations.

We sincerely thank the reviewer for the metabolic flux analysis insight to further characterize strains 26R and 28R. We have now measured all the intermediates of the MVA pathway in the two strains. According to lines 277-285, Table 1, and Fig. 4f of the revised manuscript, although with differently regulated gene expressions, the concentrations of the intermediates were comparable, except for mevalonic acid and mevalonate diphosphate. This result suggested that the sole control of transcriptional regulation in the selected colonies was likely reaching the limit, switching to the combinational control of the gene expression and the flux control.

It is also possible that the selection of 50 colonies through random colony picking may not be able to select the optimal expression of the mevalonate pathway. We have now added an experiment screening 500 colonies of the heme producing strains, which gained us several new insights of the heme pathway. The two case studies demonstrated that flux control is distributed over many enzymes, and Matrix Regulation exhibits the strength to fine-tune a given pathway, when combining with machine learning of thousands of colonies. The relevant analysis has been added to the revised manuscript in lines 286-313 and lines 350-353.

[Minor comments]

1. Line 92: Replace “knockouts employs a” with “knockouts which employs a”
2. Line 101: Replace “with distinct four nucleotides at the 3’ end, including” with “with four distinct nucleotides at the 3’ end, namely”
3. Line 134: Replace “use of protein” with “use of a protein”
4. Line 136: Replace “use” with “used”
5. Line 149: Replace “5% genes” with “5% of the genes”
6. Line 211: Replace “mis-up-regulation” with “aberrant up-regulation”
7. Line 256: Replace “ these knowledges for most pathways are still missing” with “this knowledge is still missing for most pathways”
8. Line 308: add a reference for the transformation method.
9. Line 369: Replace “revers” with “reverse”
10. Line 478: Replace “on” with “no”

We thank the reviewer for kindly pointing these out. We feel deeply sorry for the grammatical errors and typos in the manuscript. The manuscript has now been carefully revised accordingly.